# Magnetic order in 2D antiferromagnets revealed by spontaneous anisotropic magnetostriction

Maurits J. A. Houmes [1,4] ✉, Gabriele Baglioni[1,4], Makars Šiškins [1,4], Martin Lee[1], Dorye L. Esteras[2], Alberto M. Ruiz[2], Samuel Mañas-Valero [1,2], Carla Boix-Constant [2], Jose J. Baldoví [2], Eugenio Coronado [2], Yaroslav M. Blanter[1], Peter G. Steeneken [1,3] & Herre S. J. van der Zant [1]

The temperature dependent order parameter provides important information on the nature of magnetism. Using traditional methods to study this parameter in two-dimensional (2D) magnets remains difficult, however, particularly for insulating antiferromagnetic (AF) compounds. Here, we show that its temperature dependence in AF MPS$_3$ (M(II) = Fe, Co, Ni) can be probed via the anisotropy in the resonance frequency of rectangular membranes, mediated by a combination of anisotropic magnetostriction and spontaneous staggered magnetization. Density functional calculations followed by a derived orbital-resolved magnetic exchange analysis confirm and unravel the microscopic origin of this magnetization-induced anisotropic strain. We further show that the temperature and thickness dependent order parameter allows to deduce the material's critical exponents characterising magnetic order. Nanomechanical sensing of magnetic order thus provides a future platform to investigate 2D magnetism down to the single-layer limit.

Layered two-dimensional (2D) magnetic materials offer an emerging platform for fundamental studies of magnetism in the 2D limit. Their stackability into van der Waals heterostructures opens pathways to non-trivial magnetic phases and technological applications, including sensors, memories and spintronic logic devices[1]. In addition to ferromagnetism, first observed in CrI$_3$[2] and Cr$_2$Ge$_2$Te$_6$[3], antiferromagnetism in 2D materials has also been studied in FePS$_3$[4] and CrSBr[5]. Antiferromagnetic (AF) materials are of particular technological interest due to their high spin-wave propagation speed and lack of macroscopic stray fields, making them strong candidates for spintronic and magnonic applications[6–10].

For insulating, thin AF materials, such as MPS$_3$ (M(II) = Fe, Co, Ni), few methods are available to study their intrinsic magnetism. Conventional techniques, such as neutron scattering, magnetization measurement by a superconducting quantum interference device (SQUID) or vibrating sample magnetometry are challenging, due to the small volumes of exfoliated 2D materials. Other methods, suited to 2D materials, require electrical conductance, the presence of specific optical modes or ferromagnetic order; they are therefore difficult to apply[1]. In contrast, strain applied to 2D magnetic materials was shown to be extremely powerful, inducing magnetization reversal[11], reorientating the easy-axis[12], or reversing the exchange interaction[13]. In addition, the direct coupling between strain, resonance frequency and magnetization in membranes of 2D magnets, makes nanomechanical resonance a sensitive method for studying their phase transitions[14–16].

Here, we show, guided by density functional theory (DFT), that the magnetic order parameter of MPS$_3$ AF membranes can be quantified through the anisotropy in their magneto-elastic response; from its temperature dependence the critical exponents are determined, and their thickness dependence is investigated.

[1]Kavli Institute of Nanoscience, Delft University of Technology, Lorentzweg 1, 2628 CJ Delft, The Netherlands. [2]Instituto de Ciencia Molecular (ICMol), Universitat de València, c/Catedrático José Beltrán 2, 46980 Paterna, Spain. [3]Department of Precision and Microsystems Engineering, Delft University of Technology, Mekelweg 2, 2628 CD Delft, The Netherlands. [4]These authors contributed equally: Maurits J. A. Houmes, Gabriele Baglioni, Makars Šiškins. ✉e-mail: m.j.a.houmes@tudelft.nl

## Results and discussion

### First principles analysis of spontaneous magnetostriction in MPS₃

Transition-metal phosphorus trisulphides, with general formula $MPS_3$, are layered materials stacked in a monoclinic lattice with symmetry group C2/m[17], as shown in the top view of a single-layer in the paramagnetic phase, Fig. 1a, top panel. The spins of $FePS_3$ point out-of-plane, whereas both $CoPS_3$ and $NiPS_3$ are in-plane systems with their spins preferentially aligned along the $a$ axis. The intralayer AF order forms a zigzag configuration, as shown in bottom panel of Fig. 1a, leading to two opposite aligned magnetic sub-latices. The difference of the magnetization between these sub-latices is the Néel vector. In bulk $CoPS_3$ and $NiPS_3$, these layers with this staggered magnetism are stacked in a ferromagnetic (FM) fashion with Néel transition temperatures, $T_N$, around 119 and 155 K, respectively[18,19]. The interlayer magnetic interactions in $FePS_3$ are AF with a transition around 118 K[20].

To analyze the effect of magnetic ordering on the lattice, we performed first principles structural optimizations of $FePS_3$, $CoPS_3$ and $NiPS_3$ based on density functional theory (DFT). For the ground state zigzag magnetic configuration, the calculations predict a compression of the $a$ lattice parameter with respect to the crystallographic, non-magnetic structure of 2.545% and 1.328% for the Co and Fe derivatives respectively (see Table 1). In addition, the $b$ axis expands by 0.402% (Co) and 0.359% (Fe). In contrast, in $NiPS_3$ the lattice parameters remain almost unchanged. The crystal and magnetic structures are strongly connected in these compounds, which is further corroborated by simulations of different spin configurations (see Supplementary Note 1).

We studied the 2D nature of the magnetostriction by simulating the evolution of lattice parameters in multilayer monoclinic $FePS_3$ (which presents AF interlayer coupling), obtaining similar results (1.462% compression in the a axis and 0.437% expansion in the b axis). This indicates that the observed effect is independent of the stacking and interlayer interactions.

The microscopic mechanism governing the spontaneous magnetostriction in these materials is studied using orbital-resolved magnetic exchange analyses based on maximally localized Wannier functions, (see Supplementary Note 1). The analysis shows that the spontaneous magnetostriction calculated in $FePS_3$ and $CoPS_3$ arises from isotropic magnetic exchange interactions between $t_{2g}$-$t_{2g}$ orbitals. Specifically, for $FePS_3$ the main magnetic exchange channels, substantially affected by

the compression of the $a$ and expansion of the $b$ lattice parameters, are the ones involving $t_{2g}$-$t_{2g}$ interactions of FM nature. The changes in the lattice parameters result in an increase in $J_1$ and $J_1'$ due to a decrease in distance between the $d_{yz}$-$d_{yz}$ and $d_{xz}$-$d_{xz}$ orbitals, respectively (Fig. 1c). Simultaneously, these changes cause a decrease of $J_2$ due to a larger separation of the $d_{xz}$-$d_{xz}$ orbitals (Fig. 1d). This is compatible with the electron configuration of $Fe^{2+}$ ($d^6$), which has these orbitals partially filled and allows FM hopping between them (Fig. 1e, f).

This hopping effect also occurs for $Co^{2+}$ ($d^7$) although the additional electron present for Co blocks the $d_{xy}$-$d_{xy}$ pathway (Supplementary Note 1, Fig. S2). This results in a stronger effect along $J_1$ and $J_1'$ for the optimized structure, maximizing FM interactions in the zigzag chain, which involve the $d_{yz}$-$d_{yz}$ and $d_{xz}$-$d_{xz}$ orbitals, respectively. For the $Ni^{2+}$ derivative ($d^8$), the $t_{2g}$ energy levels are fully occupied (Supplementary Note 1, Fig. S3), which results in a blocking of the $t_{2g}$-$t_{2g}$ magnetic super-exchange channels. This leads to an almost negligible modification in the lattice parameters of the optimized structure with respect to the crystallographic non-magnetic one.

### Resonance frequency changes due to spontaneous magnetostrictive strain

The predicted anisotropic change of lattice parameters when going from the paramagnetic to the AF phase, causes compressive stress, $\sigma_a$, and tensile stress, $\sigma_b$, along the $a$ axis and $b$ axis respectively, as illustrated in Fig. 1a, bottom pannel. To quantify this anisotropy appearing at the phase transition, we use rectangular membranes, shown in Fig. 2b, to nanomechanically probe stress variations, along a specific crystallographic axis[21] (see Supplementary Note 2). In the following analysis, we neglect the stress contribution from the thermal expansion of the substrate, as this is small compared to that of the $MPS_3$ compounds[14].

The resonance frequency of the fundamental mode of a rectangular membrane, $f_{res}$, is approximately given by[22]:

$$f_{res} \approx \frac{1}{2}\sqrt{\frac{1}{\rho}\left[\frac{1}{w^2}\sigma_w + \frac{1}{l^2}\sigma_l\right]}, \qquad (1)$$

where $\rho$ is the mass density, $w$ and $l$ are respectively the width and length of the membrane, as indicated in Fig. 2b, and $\sigma_{w,l}$ are the stresses parallel to these directions. For high-aspect-ratio membranes ($w \ll l$),

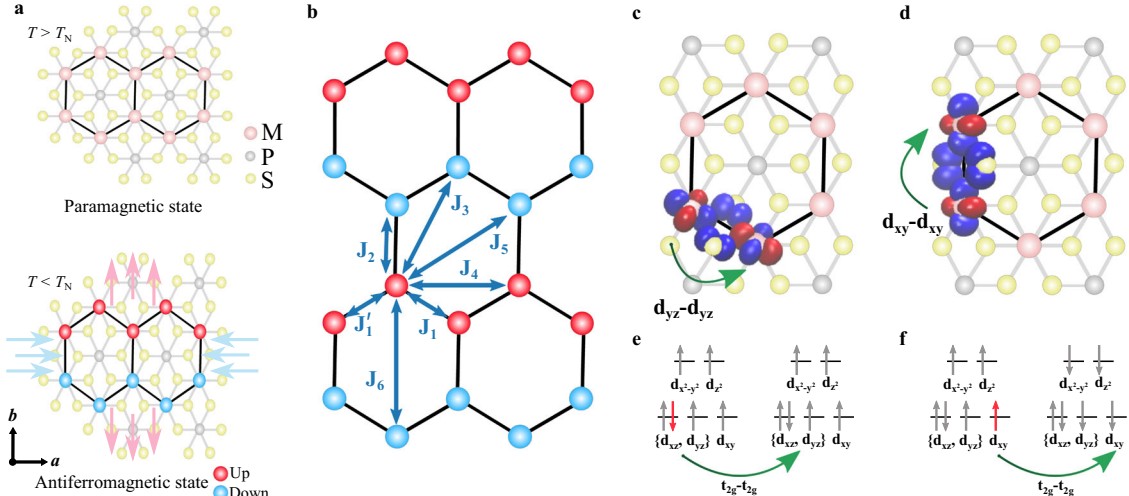

**Fig. 1 | Magnetostriction in MPS₃ membranes. a,** top panel, Crystalline structure of $MPS_3$ in the paramagnetic phase ($T > T_N$). Black hexagons indicate the organization of magnetic atoms in the lattice. **a,** bottom panel, Crystalline structure of $MPS_3$ at the AF phase ($T < T_N$) as it elongates in the $b$ and contracts in the $a$ direction. Light blue and red arrows indicate the axial lattice distortion. **b,** Illustration of the exchange interaction parameters included into the Heisenberg spin Hamiltonian.

Calculated maximally localized Wannier orbitals. Green arrows illustrate the most relevant FM superexchange channels for $J_1$ ($J_1'$) (**c**) and $J_2$ (**d**), corresponding with the $d_{yz}$-$d_{yz}$ ($d_{xz}$-$d_{xz}$) and $d_{xy}$-$d_{xy}$ orbitals, respectively. **e–f,** Electron configuration of the $Fe^{2+}$ magnetic ions connected by $J_1$ (**e**) and $J_2$ (**f**), showing parallel and antiparallel spin orientations, respectively.

the mechanical resonance frequency is mostly determined by the stress along the shortest direction, $\sigma_w$. The membranes shown in this work range in aspect ratio from 1 to 5 up to 1 to 12, corresponding to the $\sigma_l$ pre-factor being smaller by a factor of 25 up to 144 as compared to the $\sigma_w$ pre-factor.

We study the resonance frequency of thin MPS$_3$ flakes suspended over star-shaped cavities with 30° angular resolution, as shown in an example device in Fig. 2b. When the longest side of the cavity is aligned

along a crystallographic axis ($a$ or $b$) and $w \ll l$, its fundamental resonance frequency ($f_a$ or $f_b$) is determined by the stress along the perpendicular axis ($\sigma_b$ or $\sigma_a$):

$$f_a \approx \frac{1}{2}\sqrt{\frac{1}{\rho w^2}\sigma_b} \text{ and } f_b \approx \frac{1}{2}\sqrt{\frac{1}{\rho w^2}\sigma_a}. \tag{2}$$

On cavities oriented at an intermediate angle, $\theta$, (defined with respect to the $b$ axis), the resonance frequency is:

$$f_\theta(T) \approx \frac{1}{2}\sqrt{\frac{1}{\rho w^2}\left[\sigma_{a,\theta} + \sigma_{b,\theta}\right]},$$
$$\sigma_{a,\theta} = \frac{E}{(1-\nu^2)}(\cos^2\theta + \nu\sin^2\theta)(\bar{\epsilon} - \epsilon_{ms,a}), \tag{3}$$
$$\sigma_{b,\theta} = \frac{E}{(1-\nu^2)}(\sin^2\theta + \nu\cos^2\theta)(\bar{\epsilon} - \epsilon_{ms,b}),$$

where we have used the constitutive equations for a magnetostrictive membrane with plane stress[23], while only keeping the anisotropy in the

**Table 1 | CoPS$_3$, FePS$_3$ and NiPS$_3$ lattice parameters of the crystallographic non-magnetic (NM) and fully optimized zig-zag antiferromagnetic (AF-zigzag) configurations, as calculated by DFT (see Supplementary Note 1)**

|  | CoPS$_3$ | | FePS$_3$ | | NiPS$_3$ | |
|---|---|---|---|---|---|---|
| Lattice parameter (Å) | $a$ | $b$ | $a$ | $b$ | $a$ | $b$ |
| NM | 5.895 | 10.19 | 5.947 | 10.301 | 5.812 | 10.07 |
| AF-zigzag | 5.745 | 10.231 | 5.868 | 10.338 | 5.817 | 10.061 |
| Change (%) | −2.545 | +0.402 | −1.328 | +0.359 | +0.086 | −0.089 |

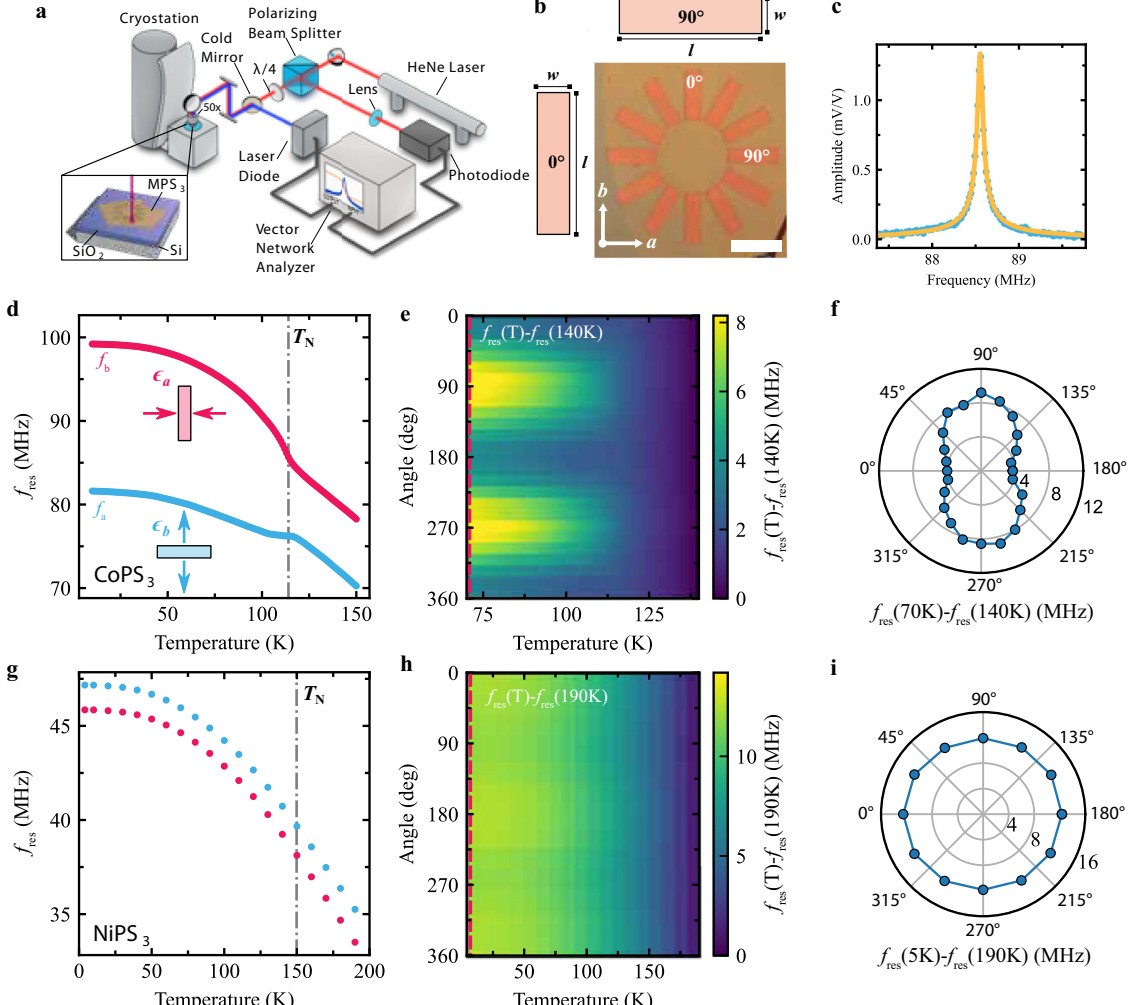

**Fig. 2 | Angle-resolved mechanical characterization via laser interferometry.** **a** Schematic illustration of the laser interferometry setup and sample with rectangular cavity array. **b** Optical image of the rectangular membranes array for a CoPS$_3$ sample. The $a$ and $b$ axis are determined from the resonance frequency behavior. Scale bar: 12 μm. Schematic of 0° and 90° membranes from the array where $w$ is the width of the membrane and $l$ its length. **c** Measured amplitude of the fundamental resonance peak in a CoPS$_3$ drum at $T = 10$ K and Lorentzian fit used to extract the fundamental resonance frequency, $f_{res}$, and quality factor, $Q$. **d** Temperature dependence of $f_{res}$ of a CoPS$_3$ rectangular membrane, shown are $f_a$ (blue) and $f_b$ (red) as defined in Eq. (2). The arrows show the dominant magnetostrictive strain contributions for the corresponding cavities. The dashed line indicates the transition temperature $T_N$ extracted from the data. **e** Resonance frequency difference, $f_{res}(T) - f_{res}(140 \text{ K})$, as a function of angle and temperature. The dashed line indicates the transition as in (**d**). **f** Polar plot of $f_{res}(T) - f_{res}(140 \text{ K})$ taken along the red dashed line in (**e**). **g**–**i** follow the same structure as (**c**–**e**) for NiPS$_3$ resonators with negligible anisotropy, measured between 5 K and 190 K.

magnetostriction coefficient, Supplementary Note 2. Here, $E$ is the Young's modulus and $\nu$ is Poisson's ratio of the material. Moreover, we have $\bar{\epsilon} = \epsilon_{fab} - \epsilon_{th}$, with $\epsilon_{fab}$ the residual fabrication strain and $\epsilon_{th}$ the phononic thermal expansion induced strain variation. The magnetostrictive strain along the $a$ and $b$-axes is given by $\epsilon_{ms,a,b} = \lambda_{a,b}L^2$, respectively (see Supplementary Note 3 for a detailed derivation of Eq. (3)), where $\lambda_{a,b}$ are magnetostriction coefficients and $L^2$ is the AF order parameter squared.

The temperature dependence of the resonance frequency comprises two contributions: one due to the phononic thermal expansion coefficient $\alpha$, given by $\epsilon_{th}(T) = \int_{T_0}^{T} \alpha(\tilde{T})\mathrm{d}\tilde{T}$, where $T_0$ is a reference temperature and $\tilde{T}$ the integration variable, and the magnetostrictive contribution $\epsilon_{ms,a,b}(T) = \lambda_{a,b}L^2(T)$. The former contribution is a slowly varying function of $T$, while the latter term contains the staggered magnetization, which increases abruptly near the phase transition; it thus can be used to determine $L(T)$, as we will show below. We assume $\lambda_{a,b}$ to be $T$ independent, as its temperature dependence will be negligible when compared to that of $L(T)$.

## Nanomechanical determination of the order parameter

To quantify the anisotropy in the magnetic membranes, a laser interferometry technique is used to measure their resonance frequency as a function of temperature[24]. A MPS$_3$ flake, suspended over holes in a patterned Si/SiO$_2$ chip, Fig. 2b, is placed inside a cryostat with optical access as shown in Fig. 2a. Both actuation and detection are done optically, by means of a power-modulated blue laser which optothermally excites the fundamental resonance, and a constant red laser

which measures the change in the reflected signal resulting from the membrane's motion[14]. A typical resonance is shown in Fig. 2c, along with the damped harmonic oscillator model fit defining the resonance frequency. Figure 2d shows that in CoPS$_3$ $f_a$ and $f_b$ exhibit a similar temperature dependence for $T > T_N$, while diverging behavior below the phase transition $T < T_N$ is visible, namely an increase of $f_a$ and a decrease of $f_b$ relative to an overall isotropic increase. This sudden change in $f(T)$ for the perpendicular cavities, occurring near $T_N$, constitutes, in accordance with the DFT calculations, the central result of this work as it shows that the magnetic ordering in MPS$_3$ leads to anisotropic strain and thus spontaneous magnetostriction. We further note that strictly speaking, $T_N$ should be replaced by $T_N^*$ which includes the effects of strain (see Supplementary Note 2). For simplicity, we here use the notation $T_N$ for the measured transition temperatures.

The anisotropic behavior of CoPS$_3$ in the AF state is even more evident in Fig. 2e, where $f_{res}(T) - f_{res}(140\,\mathrm{K})$ for the different cavities of the star-shaped sample are plotted as a function of $\theta$ and temperature. The polar plot in Fig. 2f shows the data along the red dashed line at $T = 70$ K in Fig. 2e and results in a characteristic dumbbell-shape. Similar anisotropic behavior is observed in FePS$_3$ as shown in Supplementary Note 4. On the contrary, for NiPS$_3$ negligible anisotropy is observed in the angle-resolved magnetostriction data in Fig. 2g–i.

To obtain $L(T)$ from the data, we first subtract the pretension contribution from the resonance frequency $f_\theta(T_0)$ by calculating $\tilde{f}_\theta^2(T) = f_\theta^2(T) - f_\theta^2(T_0)$, for each angle, where $T_0 = 150$ K is the highest temperature in our measurements. The resulting values of $\tilde{f}_\theta^2(T)$ along the crystalline axes $a$ and $b$ are shown in Fig. 3a, d, g for the three MPS$_3$

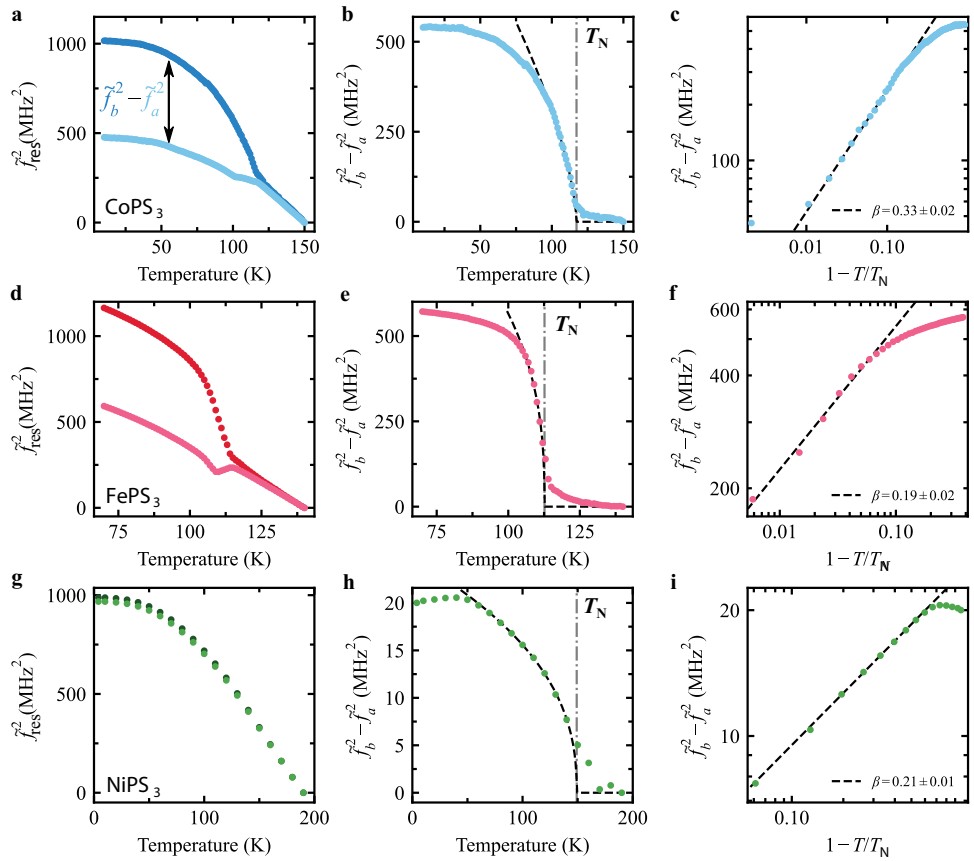

**Fig. 3 | Anisotropy and critical behavior in resonance frequency of MPS$_3$ (M(II) = Co, Fe, Ni) membranes. a** Pretension corrected resonance frequency $(\tilde{f}_a^2(T) = f_a^2(T) - f_a^2(150\,\mathrm{K})$ (light) and $\tilde{f}_b^2(T) = f_b^2(T) - f_b^2(150\,\mathrm{K})$ (dark) of rectangular membranes of CoPS$_3$. **b** Difference of the corrected frequency squared $\tilde{f}_b^2 - \tilde{f}_a^2$ proportional to the order parameter $L^2$ from Eq. (4). The dashed-dotted line indicates the measured transition temperature $T_N$. The dashed black line is a powerlaw fit through the data close to $T_N$ (see Supplementary Note 6). **c** Difference of the corrected frequency squared $\tilde{f}_b^2 - \tilde{f}_a^2$ as a function of the reduced temperature $1 - T/T_N$. The dashed black line is the fit from b where the slope defines the critical exponent $2\beta$. **d**–**f** and **g**–**i** follow the same structure as (**a**–**d**) for FePS$_3$ and NiPS$_3$ resonators, respectively.

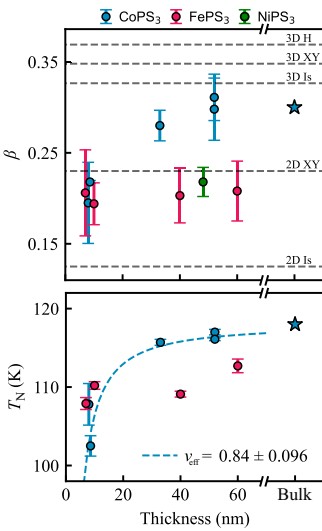

**Fig. 4 | Thickness dependence of critical behavior.** Average critical exponent, $\beta$, and critical temperature, $T_N$, of MPS$_3$ resonators plotted as a function of thickness. The blue stars indicate CoPS$_3$ bulk values from[31]. Critical parameters have been determined from power law fits to $\tilde{f}_b^2 - \tilde{f}_\theta^2$, as shown in Fig. 3b, e, h, and then taking the average value over the fit parameter for all angles $\theta \neq 0$. Error bars are calculated from standard deviation of fit results for all $\theta$. The horizontal gray dashed lines in the upper plot indicate the expected values of $\beta$ for the 3D or 2D versions of the Heisenberg (H), XY or Ising (Is) models. The blue dashed line in the lower panel indicates a fit to Eq. (6) through the CoPS$_3$ data with $\nu_{eff} = 0.84 \pm 0.13$.

compounds. Note that this is not the same data as shown in Fig. 2, but of a sample with thickness and geometry closer to that of the sample Fig. 3d, e, f, for ease of comparison. With Eq. (3), we then calculate the difference $\tilde{f}_b^2(T) - \tilde{f}_a^2(T)$ which yields

$$\tilde{f}_b^2 - \tilde{f}_a^2 = \frac{E}{4\rho w^2(1+\nu)}[\lambda_a - \lambda_b]L^2. \tag{4}$$

We can now use Eq. (4) to access the critical behavior of $L$ below $T_N$ by plotting $\tilde{f}_b^2 - \tilde{f}_a^2$ as a function of temperature. As shown in Fig. 3b, e, h, the trend presents the typical critical behavior with a non-zero order parameter appearing in the ordered state for $T < T_N$. Figure 3c, f, i shows the same critical curve as Fig. 3b, e, h respectively, plotted on a logarithmic scale against the reduced temperature $(1 - T/T_N)$. Note that the difference $\tilde{f}_b^2 - \tilde{f}_a^2$ for NiPS$_3$, is substantially smaller than that of the Fe/CoPS$_3$ membranes indicative of a weaker anisotropic magnetostrictive behavior.

The angle dependence of the resonance frequencies allows us to estimate the ratio $r_{ab} = \lambda_a/\lambda_b$ between the magnetostriction parameters, $\lambda_{a,b}$, (see Supplementary Note 3). This ratio we directly compare to DFT calculations: Experimentally, we find for FePS$_3$, $r_{ab} = -2.3 \pm 0.3$ while from the DFT calculations we estimate $r_{ab} = -3.70$. For CoPS$_3$ (taking[25] $\nu_{CoPS_3} = 0.293$), the experimental value is $-1.42 \pm 0.07$ and the DFT one $-6.33$. We conclude that although both the sign and order of magnitude of the magnetostrictive anisotropy in these compounds are well reproduced in the current work, more detailed studies will be needed to obtain full quantitative correspondence with theory.

### Thickness dependence of critical behavior

When transitioning from 3D systems to 2D systems the magnetic order is generally changed. For isotropic 2D systems the Mermin-Wagner theorem forbids magnetic order as it will be destroyed by thermal fluctuations. However for anisotropic 2D systems different magnetic orders can exist, such as Ising, XY and Kosterlitz-Thouless phases[26]. By varying the thickness, i.e., the number of layers, of the MPS$_3$

compounds we are able to investigate the transition from the bulk 3D behavior to the 2D behavior.

As follows from Landau's theory of phase transitions (see Supplementary Note 2), $L(T)$ near $T_N$ is given by

$$L^2(T) = \begin{cases} 0 & \text{if } T > T_N \\ \frac{A}{2B}(T_N - T)^{2\beta} & \text{if } T < T_N, \end{cases} \tag{5}$$

where $A$ and $B$ are constants and $\beta$ is a critical exponent representative of the magnetic order. We fit Eq. (5) to the data in Fig. 3b, e, h in the region close to $T_N$ (indicated by the black dashed line in Fig. 3b, e, h) to extract the critical exponent $\beta$ and $T_N$ for the three materials (see Supplementary Note 6 for more details on the fitting procedure). In the logarithmic plot of the critical curve the fitting of a straight line shows good agreement to the data points, consistent with the result of Eq. (5). The values for $\beta$ and $T_N$ are plotted in Fig. 4 as a function of thickness, $t$, and listed in Supplementary Note 6, Table 49.

For the weakly anisotropic NiPS$_3$, $\beta = 0.218 \pm 0.016$, comparable to the value ($\beta = 0.22 \pm 0.02$) found in ref. 27, and consistent with the expected 2D XY magnetic dimensionality ($\beta_{2DXY} = 0.233$) of NiPS$_3$[28]. For FePS$_3$ we find $\beta = 0.208 \pm 0.033$, comparable with literature values[29]. For both $\beta$ and $T_N$ no appreciable thickness dependence is observed, similar to what has previously been reported in ref. 30, where changes in the critical behavior mostly become visible in the monolayer limit.

For thicker CoPS$_3$ samples ($t = 40 - 60$ nm) we find $\beta = 0.289 \pm 0.034$ close to what is reported in literature for the bulk ($\beta_{bulk} = 0.3 \pm 0.01$[18]) and consistent with the 3D Ising model. For samples with $t < 10$ nm the measured $\beta$, on the other hand, is $0.195 \pm 0.045$, closer to $\beta_{2DXY}$ as shown in the top panel of Fig. 4. This constitutes a noticeable change in $\beta$ while going from bulk to thinner samples. Similarly, we observe for CoPS$_3$ a decrease in $T_N$ from the bulk value of 118 K down to ~100 K, similar to what was previously reported in ref. 31. We fit a power law to the dependence of $T_N$ on thickness,

$$T_N(t)/T_N^{3D} \propto 1 - (C/t)^{1/\nu_{eff}}, \tag{6}$$

where $C$ is a non-universal constant related to the interlayer coupling, and $\nu_{eff}$ is an effective critical exponent related to the correlation length[32]. Fitting the CoPS$_3$ data points with $T_N^{3D} = 118$ K[31] yields $C = 1.43 \pm 0.457$ nm and $\nu_{eff} = 0.84 \pm 0.096$. This value of $\nu_{eff}$ is intermediate between the expected values of $\nu_{eff} = 0.630$ for the 3D Ising and $\nu_{eff} = 1$ for the 2D Ising models, and indicative of a transition regime[26].

In conclusion, we provide a comprehensive analysis of the anisotropic magnetostriction effect in MPS$_3$ compounds and its implications to the dynamics of membrane made from them. DFT calculations provide a microscopic explanation for the anisotropic lattice deformation in CoPS$_3$, FePS$_3$ and NiPS$_3$ which are consistent with our measurements. We further demonstrate the relation between magnetic ordering and anisotropy in the mechanical resonance frequency of suspended MPS$_3$ resonators, providing a direct measure of the AF order parameter in absence of an external magnetic field. We observe a thickness dependence in the critical behavior of CoPS$_3$ resonators[18,31], which is absent in the case of FePS$_3$. The presented technique is of particular interest for the study of 2D magnetism given the scarcity of methods available to investigate critical phenomena of van der Waals materials in the atomically thin limit.

## Methods

### Sample fabrication

Substrates consist of thermal SiO$_2$ of 285 nm thickness, grown on highly doped (Si$^{++}$) silicon. The rectangular cavities are defined via e-beam lithography using AR-P 6200 resist. After development, the exposed SiO$_2$ areas are fully etched via reactive ion etching. The AR-P 6200 resist is stripped in PRS-3000 and the sample is cleaned in an O$_2$ plasma before stamping. The exfoliation and transfer of multi-layer

MPS$_3$ flakes is done using a polydimethylsiloxaan (PDMS) transfer method. First, MPS$_3$ crystals are exfoliated onto the PDMS through scotch tape. Selected flakes are then transferred on the star-shaped cavities in the SiO$_2$/Si substrate[33].

## Laser interferometry

Samples are mounted on a heater stage which is cooled down to 4 K using a Montana Instruments Cryostation s50 cryostat with optical access. A blue diode laser ($\lambda = 405$ nm) is used to excite the membrane optothermally via AC power-modulation from a vector network analyzer (VNA)[34]. Displacements are detected by focusing a red He-Ne laser beam ($\lambda = 632$ nm) on the cavity formed by the membrane and Si substrate. The reflected light, which is modulated by the position-dependent membrane motion, is recorded by a photodiode and processed by a phase-sensitive VNA. Laser spot size is ~1 μm. In order to minimize optical heating effects the laser power is minimized, while still being able to detect the resonance. To asses the effect of optical heating laser power is varied, and the resulting resonance shift is compared to a resonance shift from a temperature variation. The frequency shift due to optical heating is found to be well bellow a shift corresponding with a 1° temperature change.

## DFT calculations

First principles spin-polarized DFT calculations in the plane wave formalism are performed as implemented in the Quantum ESPRESSO package[35]. The exchange-correlation energy is calculated using the generalized gradient approximation using the Perdew–Burke–Ernzerhof functional[36] and standard Ultra-soft (USPP) solid-state pseudopotentials. The electronic wave functions are expanded with well-converged kinetic energy cut-offs for the wave functions (charge density) of 75 (800), 85 (800), and 85 (800) Ry for Fe, Co and Ni, respectively. The crystal structures are fully optimized using the Broyden-Fletcher-Goldfarb-Shanno (BFGS) algorithm[37] until the forces on each atom are smaller than $1 \times 10^{-3}$ Ry/au and the energy difference between two consecutive relaxation steps is less than $1 \times 10^{-4}$ Ry. In order to avoid unphysical interactions between images along the non-periodic direction, we add a vacuum of 18 Å in the $z$ direction for the monolayer calculations. The Brillouin zone is sampled by a fine Γ-centered $5 \times 5 \times 1$ $k$-point Monkhorst-Pack[38]. For the simulation of the monoclinic structure of FePS$_3$, we simulate a unit cell with two layers to account for the AF interlayer coupling. Grimme-D3 dispersion corrections are added to account for van der Waals interactions between layers and the Brillouin zone is sampled by a fine Γ-centered $5 \times 5 \times 3$ $k$-point Monkhorst-Pack. A tight-binding Hamiltonian derived from first-principles is constructed in the base of Maximally-localized Wannier functions, as implemented in the Wannier90 code[39]. For that, we select the d orbitals of the metal center (Fe, Co, Ni) and the s and p orbitals of P and S to construct the connected subspace. Magnetic interactions are determined using the Green's function method in the TB2J software[40]. The orbital resolved analysis is performed after rotating the coordinate system of the crystal to align the metal-sulfur bonds direction of the octahedra with the cartesian axes.

## Crystal growth

Crystal growth of MPS$_3$ (M(II) = Ni, Fe, Co) is performed following a solid-state reaction inside a sealed evacuated quartz tube (pressure ~ $5 \times 10^{-5}$ mbar). I$_2$ was used as a transport agent to obtain large crystals. A three-zone furnace is used, where a tube with the material was placed in the leftmost zone. This side is then heated up to 700 °C in 3 h so that a temperature gradient of 700/650/675 °C is established. The other two zones are heated up in 24 h from room temperature to 650 °C and kept at that temperature for one day. The temperature is kept constant for 28 days and cooled down naturally. With this process crystals with a length up to several centimeters are obtained. Detailed description of the crystal growth and characterization can be found in earlier work[14].

## Data availability

All data supporting the findings of this article and its Supplementary Information will be made available upon request to the authors.

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

## Acknowledgements
M.Š., M.J.A.H., G.B., M.L., H.S.J.v.d.Z. and P.G.S. acknowledge funding from the European Union's Horizon 2020 research and innovation program under grant agreement number 881603. Y.M.B and H.S.J.v.d.Z. acknowledge support from Dutch National Science Foundation (NWO). D.L.E., A.M.R., S.M.-V., C.B.-C., J.J.B., E.C. acknowledge funding from the European Union (ERC AdG Mol-2D 788222, ERC StG 2D-SMARTiES 101042680 and FET OPEN SINFONIA 964396), the Spanish MCIN (Project 2DHETEROS PID2020-117152RB-100 and Excellence Unit "Maria de Maeztu"CEX2019-000919 -M), the Spanish MIU (FPU21/04195 to A.M.R.) and the Generalitat Valenciana (PROMETEO Program and APOST Grant CIAPOS/2021/215 to S.M.-V.) The computations were performed on the Tirant III cluster of the Servei d'Informática of the University of Valencia.

## Author contributions
D.L.E., A.M.R. performed the DFT and orbital resolved tight-binding calculations, supervised by J.J.B. M.Š., M.J.A.H. and G.B. performed the laser interferometry measurements and fabricated and inspected the samples. M.L. and M.J.A.H. fabricated the substrates. S.M.-V. and C.B.-C. synthesized and characterized the $FePS_3$, $CoPS_3$ and $NiPS_3$ crystals, supervised by E.C. M.Š., M.J.A.H. and G.B. analyzed the experimental data. M.Š., M.J.A.H., Y.M.B., and P.G.S. modeled the experimental data. H.S.J.v.d.Z. and P.G.S. supervised the project. The paper was jointly written by all authors with a main contribution from M.J.A.H. All authors discussed the results and commented on the paper.

## Competing interests
The authors declare no competing interests.
