## [Peer Review File · Nature Communications]

Reviewers' Comments:

Reviewer #1:

None

Reviewer #2:

Remarks to the Author:

Magnetism in two-dimensional materials is a topic of considerable interest to a broad and diverse community of scientists, yet one that remains challenging to study in the lab. Nanomechanical resonators based on suspended membranes of these materials offer the (rather unique) possibility to address this topic from the effect caused by magnetic ordering on the mechanical properties of the membranes. For example, the temperature dependence of the resonant frequencies of vibrational modes may reveal a change in strain caused by a structural phase transition. Some of the most interesting situations are those where such a phase transition is caused by an ordering of the magnetic moments of atomic orbitals, a phenomenon called magnetostriction.

The manuscript by Houmes and coworkers is about the study, in 3 different two-dimensional compounds, of an antiferromagnetic ordering accompanied by an anisotropic change in strain. The novelty of this important work, in my opinion, is threefold. (i) A microscopic model is presented that explains the origin of the magnetic phase transition, and that is later found to account for the measured data. (ii) A scheme to measure the anisotropic mechanical response is devised, based on which meaningful physical quantities are measured. (iii) A significant change in resonant frequency near the Neel temperature for the fundamental mode of devices whose crystal a-axis is oriented along the length of the cavity reveals an anisotropic change in strain, which is understood to be caused by an antiferromagnetic transition.

The highlight of this work is the demonstration that nanomechanical resonators are ideally suited for studying magnetic anisotropies in low-dimensional systems. As clearly explained in the main text and further detailed in Supplementary Information, measuring the resonant frequency of resonators whose crystal lattice is oriented along different directions with respect to the cavity makes it possible to access the magnetic order parameter (which is the magnitude of the Neel vector) almost directly. Using this approach, a power law of the magnetic order parameter is found over one order of magnitude in temperature, with an exponent that depends on the resonator material and that is consistent with existing results.

The manuscript is very well written. Measurements are of high quality, and the configuration to probe mechanical anisotropies is ingenious. The analysis is very thorough. The derivations of the anisotropic resonant frequency and of the interplay of magnetostriction and vibrations will be useful to researchers. I found the answers to all my questions in the text. As a minor point, the authors may want to scrutinize the text around Eq. 40 and the equation itself in Supplementary Information for possible typos.

Reviewer #3:

Remarks to the Author:

Maurits J. A. Houmes et. al. report magnetic order in the 2D magnetic van der Waal MPS3 (M = Fe, Co, Ni) compounds via spontaneous anisotropic magnetostriction measurements. The role of orbital-driven magnetic exchange in the anisotropic strain has been studied using DFT calculations. The article presents a new method for detecting magnetostriction in 2D magnetic vdW materials, which can identify a magnetic ordering transition temperature without applying magnetic field. In this sense, it's a novel approach, but it relies on the strong connection between the magnetostriction and the spontaneous magnetization in the MPS3 compounds. Therefore, the applicability of the demonstrated technique is limited to only those materials which have similar properties. This is clear from the fact that NiPS3 in Figures 3g&h does not show any anomalies at the Neel transition, consistent with the absence of magnetostriction at the ordering temperature. The presence of magnetostriction in the Fe and Co compounds has already been reported (<https://journals.aps.org/prb/abstract/10.1103/PhysRevB.101.024415>) and this work does not further elucidate the nature of the interactions. While the properties of the MPS3 compounds

provide a nice demonstration case to nanomechanically sense magnetic order, this work does not significantly advance our understanding of the material class. In addition to the limited applicability of the method, I believe that this work is not of high enough impact to qualify it for publication in Nature communications. I recommend it to be resubmitted to a journal better suited to highlight the technical advantages of the method, which in my opinion, represent the novelty of this work.

Specific points that could be improved for resubmission are the following:

- The article requires a thorough English proofreading.
- More explanation related to the experimental setup would be appreciated. In particular, how the material is mounted onto the cavity (e.g. how are the flakes attached to the cavity, with glue or electrostatically?).
- It would be useful to include more discussion about other effects that may be influencing the resonance frequency (optical heating, magnetic torque/magnetotropic, mismatch in thermal strains between substrate and sample)
- How "approximate" are equations 2? At what length scale does σ_a for example contribute to f_a , and how far away are you from this $w \ll l$ limit?
- "Opposite" behavior is used to describe Figure 2d. It should be specified that this is true only JUST below T_n . Please also elaborate on the overall stiffening below T_n in both. For example, it's decreasing for the b-axis only if the behavior above T_n is fit and extrapolated down to lowest temperatures and then subtracted from the data.
- The "long side of cavity is crystal direction a and has fundamental frequency f_a , which is determined by σ_b " But then I don't understand how to connect Figure 2d with Figure 3a. It's not clear if the arrows in Figure 2d correspond to the strain direction or the crystal axis. Since it's a double arrow, I would have guessed that it's describing strain, but then I would have thought that the upper curve in Figure 3a is f_a , not f_b as labelled.
- Figure 3 caption – there is no "red" and "blue" in the plot for panel a. Also, the numbers don't seem to work out. For example, f_a in figure 3a does not appear to be the value one gets from Figure 2d subtracted from the 150K data and then squared.
- The experimentally-obtained magnetostriction coefficients do not match well with DFT. Other effects must be considered.
- It seems that the assigned a- & b-axes are based on the crystal morphology. Has the relation of this with the crystallographic a- and b-axis been checked experimentally?
- Supplementary Fig. 9ad, the membrane length and width are quite different for CoPS3 and FePS3. What is the reason for this difference?
- The studied materials are susceptible to strain and bending during handling. Has this been accounted for?
- It has been reported that FePS3 has a broken mirror symmetry in the ac-plane (<https://iopscience.iop.org/article/10.1088/2053-1583/abed3/meta>) that leads to the inclusion of off-diagonal components in the magnetic susceptibility. How does the monoclinic structure of FePS3 affect the relative magnetostriction in the a- versus b-direction? Is this considered?
- The authors observe a thickness dependence in the critical behavior of the Co analog, but the relevance of this is not clear.

Reviewer #2 (Remarks to the Author):

Magnetism in two-dimensional materials is a topic of considerable interest to a broad and diverse community of scientists, yet one that remains challenging to study in the lab. Nanomechanical resonators based on suspended membranes of these materials offer the (rather unique) possibility to address this topic from the effect caused by magnetic ordering on the mechanical properties of the membranes. For example, the temperature dependence of the resonant frequencies of vibrational modes may reveal a change in strain caused by a structural phase transition. Some of the most interesting situations are those where such a phase transition is caused by an ordering of the magnetic moments of atomic orbitals, a phenomenon called magnetostriction.

The manuscript by Houmes and coworkers is about the study, in 3 different two-dimensional compounds, of an antiferromagnetic ordering accompanied by an anisotropic change in strain. The novelty of this important work, in my opinion, is threefold. (i) A microscopic model is presented that explains the origin of the magnetic phase transition, and that is later found to account for the measured data. (ii) A scheme to measure the anisotropic mechanical response is devised, based on which meaningful physical quantities are measured. (iii) A significant change in resonant frequency near the Neel temperature for the fundamental mode of devices whose crystal a-axis is oriented along the length of the cavity reveals an anisotropic change in strain, which is understood to be caused by an antiferromagnetic transition.

The highlight of this work is the demonstration that nanomechanical resonators are ideally suited for studying magnetic anisotropies in low-dimensional systems. As clearly explained in the main text and further detailed in Supplementary Information, measuring the resonant frequency of resonators whose crystal lattice is oriented along different directions with respect to the cavity makes it possible to access the magnetic order parameter (which is the magnitude of the Neel vector) almost directly. Using this approach, a power law of the magnetic order parameter is found over one order of magnitude in temperature, with an exponent that depends on the resonator material and that is consistent with existing results.

The manuscript is very well written. Measurements are of high quality, and the configuration to probe mechanical anisotropies is ingenious. The analysis is very thorough. The derivations of the anisotropic resonant frequency and of the interplay of magnetostriction and vibrations will be useful to researchers. I found the answers to all my questions in the text. As a minor point, the authors may want to scrutinise the text around Eq. 40 and the equation itself in Supplementary Information for possible typos.

We thank the Reviewer 2 for his/her positive assessment and expressing interest in our work, while also highlighting the importance of the main results. As kindly suggested, we have corrected the mistake in Supplementary equation 40 and the preceding text.

Reviewer #3 (Remarks to the Author):

Maurits J. A. Houmes et. al. report magnetic order in the 2D magnetic van der Waal MPS3 (M = Fe, Co, Ni) compounds via spontaneous anisotropic magnetostriction measurements. The role of orbital-driven magnetic exchange in the anisotropic strain has been studied using DFT calculations. The article presents a new method for detecting magnetostriction in 2D magnetic vdW materials, which can identify a magnetic ordering transition temperature without applying magnetic field. In this sense, it's a novel approach, but it relies on the strong connection between the magnetostriction and the spontaneous magnetization in the MPS3 compounds. Therefore, the applicability of the demonstrated technique is limited to only those materials which have similar properties. This is clear from the fact that NiPS₃ in Figures 3g&h does not show any anomalies at the Neel transition, consistent with the absence of magnetostriction at the ordering temperature.

The presence of magnetostriction in the Fe and Co compounds has already been reported (<https://journals.aps.org/prb/abstract/10.1103/PhysRevB.101.024415>) and this work does not further elucidate the nature of the interactions. While the properties of the MPS3 compounds provide a nice demonstration case to nanomechanically sense magnetic order, this work does not significantly advance our understanding of the material class. In addition to the limited applicability of the method, I believe that this work is not of high enough impact to qualify it for publication in Nature communications. I recommend it to be resubmitted to a journal better suited to highlight the technical advantages of the method, which in my opinion, represent the novelty of this work.

We thank the Reviewer 3 for his/her critical assessment of our work, while at the same time highlighting the novelty of the method presented. However, we disagree with the reviewer's assessment of the applicability. Hence, we would like to further clarify the key features and the matter of applicability of the method.

Indeed, we do provide a microscopic magnetostriction model that explains the origin of the magnetic phase transition and accounts for the experimental data. However, the main novelty of this work is a nanomechanical route for detection of magnetic order in antiferromagnets, by using the magnetostrictive effect, as also highlighted by the Reviewer 2. Probing of magnetic order in ultrathin low-dimensional spin systems, is very challenging with more conventional probes, including those specified in the article suggested by the Reviewer 3 [*Phys. Rev. B* 101, 024415 (2020)], especially approaching the two-dimensional limit of insulating antiferromagnets [*Nat. Nanotechnol.* 14, 408–419 (2019)]. Therefore, the methodology can provide information that is very difficult to obtain with other techniques.

We do agree with the Reviewer 3 that our nanomechanical method does require the presence of magnetostriction. Yet even in material systems where the effect is expected to be very weak our method still works. As it relies on the anisotropy of the magnetostriction rather than its strength. This is exemplified with our analysis on NiPS₃: Both the anisotropy and strength of the magnetostriction are small, which we purposefully demonstrated in figure 2g&h&i of the main text. We are however still able to extract the magnetic ordering parameters for NiPS₃ as shown in figure 3h&i of the main text, which are well-consistent with the literature [*Sci. Adv.* 7, eabf3096 (2021) and *Nat. Commun.* 10, 345 (2019)].

Therefore, the methodology is widely applicable to study magnetism in low-dimensional materials and heterostructures. Since this is an emerging, rapidly growing field, with recent papers like <https://www.nature.com/articles/s41565-019-0438-6> being cited over 1000 times, we believe this work will be of high impact and of interest to a large audience.

Specific points that could be improved for resubmission are the following:

- The article requires a thorough English proofreading.

We thank the reviewer for his/her feedback. We performed an additional round of proofreading and corrected a number of typos.

- More explanation related to the experimental setup would be appreciated. In particular, how the material is mounted onto the cavity (e.g. how are the flakes attached to the cavity, with glue or electrostatically?).

We thank the reviewer for his/her question. The 2D materials are exfoliated using the scotch tape method, [1], and transferred using a standard PDMS transfer technique. [2] This is a standard technique for fabrication of a wide range of 2D material membranes [4]. Due to the high surface area to volume ratio van der Waals forces provide enough adhesion to the substrate, which is well known in literature. [3]

We have added the relevant reference to the Methods section of the main text to clarify this for the reader.

[1] K. S. Novoselov et al. *Science* **2004**, 306, 666.

[2] Andres Castellanos-Gomez et al. *2D Mater.* **2014**, 1, 011002.

[3] Koenig et al. *Nat. Nanotechnol.* **2011**, 6, 543.

[4] Hu, Z. et al. *Chin. J. Chem.* **2020**, 38(9), 981-955.

- It would be useful to include more discussion about other effects that may be influencing the resonance frequency (optical heating, magnetic torque/magnetotropic, mismatch in thermal strains between substrate and sample)

We thank the reviewer for his/her suggestions. We have added a discussion of optical heating to the method section. We neglect the thermal strain mismatch between substrate and sample as the contribution from the substrate is small, as discussed in reference [14] of the main text. The effect is isotropic so it cancels out when taking the difference $f_b^2 - f_a^2$, meaning it would not affect our main results. In our system no external magnetic field is applied therefore no magnetotropic effects are generated [Phys. Rev. B 108, 035111 (2023)]. However, we thank the reviewer for sharing his/her insightful ideas, since this will likely be relevant for further studies using this method in combination with an applied external field.

- How “approximate” are equations 2? At what length scale does σ_a for example contribute to f_a , and how far away are you from this $w \ll l$ limit?

We thank the reviewer for his/her question. The aspect ratio for all cavities in this study is above 1:5 for the width:length. Substituting this ratio into equation 1 we can determine the relative contributions between σ_w and σ_l to f_{res} as follows: We substitute l for $5*w$ into equation 1 and find that the pre-factor for σ_l is only 4% of the pre-factor of σ_w , corresponding to suppression of the

effect of longitudinal stress by a factor of 25 or more (up to a factor 144 for the largest studied aspect ratio of 1 : 12). We have further clarified this on page 3 of the main text by the addition of the following:

“The membranes shown in this work range in aspect ratio from 1 to 5 up to 1 to 12, corresponding to the σ_l pre-factor being smaller by a factor of 25 up to 144 as compared to the σ_w pre-factor.”

- “Opposite” behaviour is used to describe Figure 2d. It should be specified that this is true only JUST below T_n . Please also elaborate on the overall stiffening below T_n in both. For example, it’s decreasing for the b-axis only if the behaviour above T_n is fit and extrapolated down to lowest temperatures and then subtracted from the data.

We thank the reviewer for his/her suggestion. The described behaviour persists everywhere below T_n but is most prominent just below T_n as can be seen in figure 2e of the main text. The overall stiffening is isotropic and not related to the emergence of the magnetic order and the described relative decrease is indeed relative to this isotropic stiffening. We have clarified this in the main text which now reads:

“Figure 2d shows that in CoPS3 f_a and f_b exhibit a similar temperature dependence for $T > T_n$, while diverging behaviour below the phase transition $T < T_n$ is visible, namely an increase of f_a and a decrease of f_b relative to an overall isotropic increase.”

- The “long side of the cavity is crystal direction a and has fundamental frequency f_a , which is determined by σ_b ” But then I don’t understand how to connect Figure 2d with Figure 3a. It’s not clear if the arrows in Figure 2d correspond to the strain direction or the crystal axis. Since it’s a double arrow, I would have guessed that it’s describing strain, but then I would have thought that the upper curve in Figure 3a is f_a , not f_b as labelled.

The reviewer’s remark is correct. We corrected and clarified the labelling in Figure 2d. We have also adjusted the caption, which now reads:

“Temperature dependence of f_{res} of a CoPS₃ rectangular membrane, shown are f_a (blue) and f_b (red) as defined in Eq. (2). The arrows show the dominant magnetostrictive strain contributions for the corresponding cavities. The dashed line indicates the transition temperature T_N extracted from the data.”

- Figure 3 caption – there is no “red” and “blue” in the plot for panel a.

We thank the reviewer for his/her suggestion and have corrected the caption:

“Pretension corrected resonance frequency ($\tilde{f}_a^2(T) = f_a^2(T) - f_a^2(150K)$ (light) and $\tilde{f}_b^2(T) = f_b^2(T) - f_b^2(150K)$ (dark)) of rectangular membranes of CoPS₃.”

Also, the numbers don’t seem to work out. For example, f_a in figure 3a does not appear to be the value one gets from Figure 2d subtracted from the 150K data and then squared.

The reviewer is correct. However, this is not a mistake. The data shown in figure 3a is of a different sample compared to that of figure 2d in order to present samples of similar thickness and cavity ratios next to

each other for comparative reasons. This has been clarified in page 5 of the main text to avoid any confusion of the reader.

- The experimentally-obtained magnetostriction coefficients do not match well with DFT. Other effects must be considered.

We thank the reviewer for expressing his/her critical assessment. The DFT agrees in sign and order of magnitude but indeed is still different from the experimentally obtained value, as we highlighted in the main text. Even though the exact correspondence between DFT calculations and experimental results is a challenging task, a comparable order of magnitude of the effect allowed us to get a qualitative understanding of origins of the effect, which we believe to be related to isotropic magnetic exchange interactions between t_{2g} - t_{2g} orbitals. Full quantitative correspondence would require more detailed studies and including more complex additional potentials, which lies beyond the scope of this work.

- It seems that the assigned a- & b-axes are based on the crystal morphology. Has the relation of this with the crystallographic a- and b-axis been checked experimentally?

The assignment of the a- and b-axis was done based on the behaviour of the membranes - not the crystals' morphology. We first determine the angled cavities with the membrane which contracts and the one that expands. Then, we label these according to the behaviour expected from the DFT calculation and literature. An independent way to verify the crystallographic axis would be through cryogenic XRD, but this is beyond the capabilities of our lab.

- Supplementary Fig. 9ad, the membrane length and width are quite different for CoPS3 and FePS3. What is the reason for this difference?

We thank the reviewer for his/her question. The reason for this is a difference in ease of fabrication and use. The presence of the effect does not depend on the overall size of the cavity. We therefore studied different membrane sizes, guided by what was allowed by the fabrication and the lateral flake size.

- The studied materials are susceptible to strain and bending during handling. Has this been accounted for?

We thank the reviewer for his/her question. The strain and bending induced during the fabrication are the reasons for the offset @150K in figures 2d and 2g. To account for this we introduce the relative normalisation in figure 3 ($\sim f^2(T) = f^2(T) - f^2(150K)$).

- It has been reported that FePS3 has a broken mirror symmetry in the ac-plane (<https://iopscience.iop.org/article/10.1088/2053-1583/abed3/meta>) that leads to the inclusion of off-diagonal components in the magnetic susceptibility. How does the monoclinic structure of FePS3 affect the relative magnetostriction in the a- versus b-direction? Is this considered?

We thank the reviewer for his/her observation. We performed the structural optimization of multilayer FePS₃ monoclinic structure, which includes the broken mirror symmetry, considering AF interlayer coupling. After the optimization of the bulk crystal structure, the lattice parameters agree with the results of Table 1 of the main text, which was obtained for monolayer FePS₃. In particular, we can see that the a lattice parameter is compressed from $a = 5.947 \text{ \AA}$ in the non-magnetic state (NM) to $a = 5.860 \text{ \AA}$ in the antiferromagnetic state (AF-zigzag), while b is expanded from $b = 10.301 \text{ \AA}$ (NM) to $b = 10.346 \text{ \AA}$ (AF-zigzag). This shows that the spontaneous magnetostriction is 2D in nature.

We now include in on page 2 of the main text the following explanation that clarifies the 2D nature of the observed effect:

“We studied the 2D nature of magnetostriction by simulating the evolution of lattice parameters in monoclinic FePS₃ (which presents AF interlayer coupling), obtaining similar results (1.462% compression in a and 0.437% expansion in b). This indicates that the observed effect is independent of the stacking and interlayer interactions.”

Also, our experiments measure $\epsilon_{ms,a}$ and $\epsilon_{ms,b}$ which are the result of the combined effect of both the staggered magnetization vector and the magnetostriction matrix, meaning we measure the combined effect of the diagonal and off-diagonal components. The monoclinic structure of FePS₃ can induce off-diagonal elements to the magnetostriction matrix, but to isolate their contribution requires an independent measure of the staggered magnetization vector which is beyond the scope of this work.

- The authors observe a thickness dependence in the critical behaviour of the Co analog, but the relevance of this is not clear.

We thank the reviewer for pointing out that the relevance of this part of the study was not clear to the reader.

While magnetic phase transitions can always occur at a finite temperature in 3D systems, they become more complex in 2D: In a truly 2D case, the existence of magnetic long-range order at any finite temperature crucially depends on spin dimensionality and is determined by the physical parameters of the system, for example, the presence and strength of magnetic anisotropy, which can significantly influence their behaviour.

By studying the change in critical behaviour, and thus anisotropy and spin dimensionality with CoPS₃ thickness, we gain insights into the conditions required for the emergence or suppression of long-range magnetic order in 2D XY systems as well as at a crossover between 3D and 2D. We show that even for a 8 nm of membrane thickness, ~10 layers of the 2D material, we already (quite remarkably) observe alteration of long-range ordering towards the 2D XY case. Our studies thus help to understand the ordering mechanisms in this system as well as renormalization of magnetic anisotropy better in reduced dimensions, as also pointed out by the Reviewer 2.

We now added the following clarifying passage to the beginning of the section “Thickness dependence of critical behaviour” of the main text:

“When transitioning from 3D systems to 2D systems the magnetic order is generally changed. For isotropic 2D systems the Mermin-Wagner theorem forbids magnetic order as it will be destroyed by thermal fluctuations. However, for anisotropic 2D systems different magnetic orders can exist, such as Ising, XY and KT phases. [27] By varying the thickness, i.e., the number of layers of the MPS₃ compounds we are able to investigate the transition from the bulk 3D behaviour to the 2D behaviour.”

Reviewers' Comments:

Reviewer #3:

Remarks to the Author:

The revised version of the manuscript, titled "Revelation of Magnetic Order in 2D Antiferromagnets Through Spontaneous Anisotropic Magnetostriction" submitted by Maurits J. A. Houmes et. al., commendably reflects the authors' diligent response to various concerns previously raised in the review process. The revised manuscript has indeed shown improvement in several sections, including changes in the introduction part, results and discussion, and Fig 2 and Fig 3 of the main manuscript.

However, despite these positive changes, there are still some sections of the manuscript that require further clarification and revisions. I note that some of the queries and suggestions highlighted in the initial referee report have not been fully addressed in the revised version. In their response, the authors stated that certain comments were either beyond the scope of the study or that they lacked the relevant characterization facility to address them. While these limitations are understandable, it is essential to ensure that the manuscript meets the required standards for publication in Nature Communications.

I would like to draw attention to the fact that, in its current form, the manuscript still does not fully meet the necessary criteria for publication in Nature Communications. Some issues that require additional attention include the overall advancement in terms of the materials class and scientific questions and the general applicability of the technique to the general class of 2D materials. The article is still focused on the technique and provides many important details regarding the experimental setup.

In summary, while the revised version of the manuscript has shown improvement, the response to some queries in the previous referee report indicates that not all concerns were fully addressed, and the manuscript does not currently meet the required criteria for publication in Nature Communications.

REVIEWERS' COMMENTS

Reviewer #3 (Remarks to the Author):

The revised version of the manuscript, titled "Revelation of Magnetic Order in 2D Antiferromagnets Through Spontaneous Anisotropic Magnetostriction" submitted by Maurits J. A. Houmes et. al., commendably reflects the authors' diligent response to various concerns previously raised in the review process. The revised manuscript has indeed shown improvement in several sections, including changes in the introduction part, results and discussion, and Fig 2 and Fig 3 of the main manuscript. However, despite these positive changes, there are still some sections of the manuscript that require further clarification and revisions. I note that some of the queries and suggestions highlighted in the initial referee report have not been fully addressed in the revised version. In their response, the authors stated that certain comments were either beyond the scope of the study or that they lacked the relevant characterization facility to address them. While these limitations are understandable, it is essential to ensure that the manuscript meets the required standards for publication in Nature Communications.

I would like to draw attention to the fact that, in its current form, the manuscript still does not fully meet the necessary criteria for publication in Nature Communications. Some issues that require additional attention include the overall advancement in terms of the materials class and scientific questions and the general applicability of the technique to the general class of 2D materials. The article is still focused on the technique and provides many important details regarding the experimental setup. In summary, while the revised version of the manuscript has shown improvement, the response to some queries in the previous referee report indicates that not all concerns were fully addressed, and the manuscript does not currently meet the required criteria for publication in Nature Communications.

We thank the reviewer for the comments and kind words. Also, we appreciate that the reviewer is positive about the changes we have made to improve the manuscript based on the input given. The reviewer asks for further revisions based on the previous review. However, based on this current review we cannot further improve the manuscript, since we have already addressed the points of this reviewer raised in the previous review in our previous reply and no new or specific points are raised. Nevertheless, there were several nice suggestions proposed by the reviewer in the previous review that we will consider for future studies.